# Uncovering Signals from the Coronavirus Genome

**DOI:** 10.3390/genes12070973

**Published:** 2021-06-25

**Authors:** Enrique Canessa

**Affiliations:** The Abdus Salam International Centre for Theoretical Physics (ICTP), Science Dissemination Unit (SDU), 34151 Trieste, Italy; canessae@ictp.it

**Keywords:** SARS-CoV-2, sequence analysis, comparative genomics variants, alternating series

## Abstract

A signal analysis of the complete genome sequenced for coronavirus variants of concern—B.1.1.7 (Alpha), B.1.135 (Beta) and P1 (Gamma)—and coronavirus variants of interest—B.1.429–B.1.427 (Epsilon) and B.1.525 (Eta)—is presented using open GISAID data. We deal with a certain new type of finite alternating sum series having independently distributed terms associated with binary (0,1) indicators for the nucleotide bases. Our method provides additional information to conventional similarity comparisons via alignment methods and Fourier Power Spectrum approaches. It leads to uncover distinctive patterns regarding the intrinsic data organization of complete genomics sequences according to its progression along the nucleotide bases position. The present new method could be useful for the bioinformatics surveillance and dynamics of coronavirus genome variants.

## 1. Introduction

Chinese scientists were the first to sequence the complete genome of SARS-CoV-2 coronavirus in humans and shared their data with the rest of the world in early 2020 [1,2,3,4]. The virus presented a unique lineage for almost half of its genome, with few genetic relationships to other known viruses, especially in the genomics region encoding the spike (S-protein) responsible for the virus entry into the human host cells [5]. The race to find immunity from this SARS-CoV-2 started soon after, and since then, genomics sequences from around the globe have been added into open archives such as the global science GISAID initiative (www.gisaid.org (accessed on 22 June 2021)) for further research.

There are thousands primary sources available in GISAID that warrant urgent investigation on biometric analyses, comparisons and characterization of these sequences of different coronavirus lineages responsible for the ongoing pandemic. In particular, an analytical study of emerging SARS-CoV-2 variants (sharing mutations) is highly needed since some variants appear to be more persistent and contagious. This is a good reason for increasing genomics surveillance on the emerging variants by the development of new tools that can detect and catalog these strains in a timely manner [6].

Similarity between biological sequences forms the basis for determining whether there is sequence homology, as defined in terms of shared ancestry between them in the evolutionary history of life [7]. Although alignment methods represent the standard for sequence analysis, comparison and similarity, it is difficult to determine the best parameters to achieve optimal alignments. There are a numerous user-defined parameters to overcome gaps and mismatches found between sequences. In addition, the computational resources required by these methods increase considerably with respect to the length and number of sequences being aligned. By contrast, the use of a simple discrete-time Fourier transform (DTFT) allows to produce fast and highly accurate sequence similarity [8]. The DTFT maps genome sequences into four binary (0–1) indicator sequences and transforms them into a frequency domain. The distance of the total Fourier Power Spectra of the sequences is used as a similarity distance metric. These distance measures of DNA sequences are found to be nearly identical to those obtained using traditional alignment-based approaches. Hence, the assessment of an effective and accurate measure of gene sequence similarity and hierarchical clustering of simulated DNA and virus sequences can also be applied via DTFT and the mapping of genome sequences into binary (0–1) indicators.

In this work, we propose a new quantitative method for the examination of distinctive patterns of complete coronavirus genome data. We deal with a certain type of alternating finite series having terms converted to binary values (0,1) for the nucleotide bases (*A*)denine, (*C*)ytosine, (*G*)uanine and (*T*)hymine (or *U*racil RNA genome for single strand folded onto itself) according to their progression along the genome sequences. Our mapping into four binary projections of the coronavirus sequence follows previous studies on the three-base periodicity characteristic of protein-coding DNA sequences [9,10]. In this work we consider complete coronavirus sequences with *N* nucleotides on the order of 30,000 base pairs (bp) in length. This novel finite and alternating sum series of binary projections is applied to most variants of SARS-CoV-2 so far sequenced. It is shown that our approach provides additional information to conventional genomics similarity computations by alignment methods [1] and to the Fourier Power Spectrum in the ‘frequency’ domain of the associated binary sequences [11,12].

By this new method we uncover distinctive signals of the intrinsic gene organization revealed by the genome sequences of the single-stranded RNA coronaviruses. In particular, we analyze the genome sequences for coronavirus variants of concern—B.1.1.7 (Alpha), B.1.135 (Beta) and P1 (Gamma)—and coronavirus variants of interest—B.1.429–B.1.427 (Epsilon) and B.1.525 (Eta). The present new method could be useful for the surveillance and dynamics of genome variants in general. The coronavirus genome is RNA, not DNA, and the nucleotide base *T* is denoted as *U*. As for the severe acute respiratory syndrome coronavirus 2 isolate Wuhan-Hu-1 complete genome (Accession ID: MN908947) [1,2,3,4], and the genome sequences considered in this work from GISAID, the reverse transcript generation method of RNA into a double-stranded DNA copy of a messenger mRNA molecule is usually reported for best results. The empirical relationship between genes and finished proteins states that three nucleotide bases, or mRNA codons, encode an amino acid, which build up proteins. This allows to correlate symbols used for nucleic acids (polymers of nucleotides) to that of proteins (polymers of amino acids) as used in this work. The series of processes bringing genetic code from DNA to proteins, through mRNA by transcription and translation, is the “central dogma” of modern molecular biology.

## 2. Similarity and Power Spectrum Analysis

The encoded genes in a sequence of four nucleotides, represented by the symbols A,C,G and *T*, store instructions to assemble and reproduce every living organism. In DNA, nucleotides of one strand are complementary to those of the opposite strand according to the pairing rules *A*-*T* and *C*-*G*. Similarity plots of SARS-like CoVs and bat SARS-like CoVs complete sequences of A,C,G and *T* revealed apparent recombination events, useful to understand the probable coronavirus pandemic origins [1,2,3,4]. Genome sequences of SARS-CoV-2 from the city of Wuhan, in China, exhibit a high level of genetic similarity (88%) to bat-derived severe acute respiratory syndrome (SARS)-like coronaviruses: bat-SL-CoVZC45 and bat-SL-CoVZXC21. Similarity plots of this kind based on the nucleotide sequence of only the S-spike gene of bat SL-CoV WIV16 have been reported in [13].

Similarity plots based on the full-length nucleotide sequence of gene variants of coronavirus are presented. In Figure 1 we illustrate the genetic similarity plot between SARS-CoV-2 Wuhan-Hu-1 (MN908947.3), and several representative full-length genome sequences of variants known as B.1.1.7 (UK), B.1.135 (South Africa), B.1.429–B.1.427 (California), B.1.525 (Nigeria) and P1 (Brazil). These complete GISAID samples come from broad geographical regions, as indicated in each figure, and were collected during different periods of time (from 1 November 2020 to 1 March 2021). Only available sequences in FASTA format with high coverage have been considered and grouped. The accession numbers of the sequences used in the figures are listed in the Appendix A.

*Note*: As of this writing, new labels have been given by WHO for SARS-CoV-2 variants of concern: B.1.1.7 (Alpha), B.1.135 (Beta) and P1 (Gamma). New labels have also been given for variants of interest: B.1.429–B.1.427 (Epsilon) and B.1.525 (Eta). A SARS-CoV-2 isolate is a variant of interest “*if, compared to a reference isolate, its genome has mutations with established or suspected phenotypic implications”*, and is variant of concern when, through a comparative assessment, it has been demonstrated to be associated with the *“Increase in transmissibility, virulence or change in clinical disease presentation; or Decrease in effectiveness of public health and social measures or available diagnostics, vaccines, therapeutics”*.

It is important to note that in all figures illustrated below, the samples for a given country correspond to data from a given variant. For example, the difference between the 33 and 19 samples from Brazil in Figure 1, Figure 2, Figure 3, Figure 4, Figure 5 and Figure 6 means that 33 of the samples taken from GISAID correspond with the Brazilian variant P1, and the other different 19 samples are also from Brazil, but they contain the UK variant B.1.1.7.

In these calculations we installed and ran, under the Linux Ubuntu O.S., the current version 36 of the FASTA sequence comparison software, which includes lalign36. It produces multiple non-overlapping alignments for protein and DNA sequences using the Huang and Miller sim algorithm for the Waterman–Eggert algorithm (github.com/wrpearson/fasta36 (accessed on 22 June 2021)). The command line used is: lalign36−m9iquery.filelibrary.file−n−f−12−g0−E10.0−m0−m“F11fasta.output” An iterative script was written to generate the results in Figure 1 within a 1000 bp window sliding every 100 bp steps.

The small regions with discordant clustering (<2%) of the different coronaviruses isolated with the first SARS-CoV-2 Wuhan-Hu-1 sequences, shown in Figure 1, suggest that these sequences reveal extreme similarities spanning throughout the genomes, as expected. The less discordant genetic similarity is found with the California and Nigeria strains. More distant relationships are detected between the query and the sequences for the Brazil and South Africa variants, which are known to share mutations (N501Y) with the UK Variant. Specifically, the latter could be a consequence of the principal common valleys found at around the first 10,000 bp and in the S-protein region (positions 21563-25384, colored in red in the figure).

Let us analyze next the power spectrum as a function of a discrete ‘frequency’ *f* of the different coronavirus sequences with *N* nucleotides (on the order of 30,000 bp) as depicted in Figure 1. Since biological sequences are strings of symbolic α=A,G,C and *T* nucleotides, binary values can be assigned to those sequences in order to apply Discrete Fourier Transform methods. Genetic sequences generate inherent signals since they are functions of an independent variable *X*, denoting the occurrence of a particular nucleotide in position *k* of the sequence. This technique has been broadly used in the literature to search for periodicity in DNA sequences [9,10,14].

In Figure 2, we examine correlations between the strings of symbols by this Discrete Fourier Transform. The plotted Power Spectrum of the coronavirus sequence is considered as the sum of the partial spectra: $\sum_{\alpha }|S_{\alpha }{\left(f\right)|}^{2}=\left(1/N^{2}\right)\sum_{\alpha }^{N}{\left|X_{\alpha ,k}\;exp\left(2\pi ifk\right)\right|}^{2}$, with discrete frequencies f=1/N,2/N⋯

For all variants considered, our plots reveal a distinct peak above noise level at around frequency f=33.3333, which identifies base periodicity property in the core genome of coronavirus. To this end, the three-base periodicity is a distinctive property of protein-coding DNA sequences from a variety of organisms [9,10].

In the past, the two methods of similarity and power spectrum have been developed for computing different types of protein features. We have seen that when applied to the genome of coronavirus variants, both of these analyses do not give much information on variant characterizations themselves. We shall show next how an alternative method can provide additional insights to conventional studies, which may help to characterize variants of the current pandemic.

## 3. Finite and Alternating Sum Series of Virus Genome

In the present paper we deal with the simplest alternating sum of the type
(1)$$E_{\alpha }\left(X\right)=\sum_{k=1}^{N}{\left(-1\right)}^{k-1}X_{\alpha ,k}\;\;\;,$$
where the variable $X_{\alpha }$ corresponds with one of the four nucleotide bases. The individual terms $X_{k}$ are associated with binary 0 or 1 values according to its presence along the complete genome sequences. This mapping follows previous studies in [9,10]. In our method, however, the arithmetic progression in Equation (Equation 1) of the genome sequences carries positive and negative signs ${\left(-1\right)}^{k-1}$ and a finite non-zero first moment of the independently distributed variables $X_{k}$.

Analyzing genomics sequencing via Equation (Equation 1) allows one to extract unique features at each bp position with a small degree of noise variations. In Figure 3, Figure 4, Figure 5 and Figure 6 we display nucleotide bases A,C,G and *T*’s imprints for the genomics strand of prevalent coronavirus variants reported from different countries for a number of samples. As reference, we illustrate complete genome sequences of SARS-CoV-2 from the city of Wuhan in China and the bat-derived severe acute respiratory syndrome (SARS)-like coronaviruses: bat-SL-CoVZC45 and bat-SL-CoVZXC21 [1]. The altered genes in the mutant coronavirus are studied by linkage analysis and genetic crosses, which allows one to locate its position in a genome and determine if the gene is similar to those already characterized. The genes are studied further by molecular sequencing as plotted here.

Equation (Equation 1) provides an additional tool to annotate the emerging virus patterns, and it aids in their screening, comparison and classification. SARS-CoV-2 has been identified as an enveloped, single-stranded, positive-sense RNA virus with a genome material encoding 27 proteins from 14 ORFs including 15 non-structural, 8 accessory, and 4 major structural proteins [15]. In the figures, it can be seen that all variants present approximately similar mirrored behavior for the first two-thirds of the viral RNA sequence. Within the area comprising the S-protein gene sequences (drawn in red) the curves undergo strong deviations among the nucleotide bases A,C,G,T with respect to curves for the first SARS-CoV-2 Wuhan-Hu-1 sequences. Studies in [15] suggest this region encoding to be a potential target to halt the entry of SARS-CoV-2.

In particular, we note distinctive trends, especially around the nucleotide region of the S-Protein. The base sequence series for Cytosine in Figure 5 shows that the UK, South Africa, Brazil and Nigeria variants share a great part of their behavior, whereas the California and Wuhan display essentially similar patters. To some degree, all signals for Guanine in Figure 6 are essentially similar. Nevertheless, the Nigerian variant here diverges rapidly. It is also observed when considering the Adenine nucleobase results in Figure 3. On the other hand, it is worth noting that the patterns for the base sequence series for Thymine in Figure 4, and for Adenine in Figure 3, display completely different convergences between the variants: UK, South Africa and from Brazil.

The positive and negative terms in the sums in Equation (Equation 1) for our discrete variables partly cancel out, allowing the series ‘to converge’ in some variants to nearly zero values for the nucleotide A-class, as depicted in Figure 3. Most other patterns for the base sequence series seem to be of non-Cauchy sequence type. Interestingly, the observed distributions of $X_{k}$ appear to be symmetric about 0 in all figures, which may imply that the mean E(X) is zero. These comparative genomics statistical representations can offer insights of inherent data organization. Curves as in Figure 3, Figure 4, Figure 5 and Figure 6 could be powerful to targeting and identify variants evolution during the course of pandemic across the world.

## 4. Future Perspective

To recap, we can conclude the following. Our method is effective and easier to apply in protein sequence comparison. It is motivated by the need to identify genetic mechanisms involved in coronavirus spreading. The added value of the alternating sums of the type in Equation (Equation 1) is to have a distinctive function representation of naturally occurring genome sequences of the virus variants $X_{\alpha ,k}$. The starting point is a finite alternating series following measures over *N* intervals. Plus and minus signs are chosen sequentially starting with +1 by default. From the view of statistics, such a sequence is equivalent to a discrete-valued time series for statistical identification and characterization of data sets [16,17].

We have shown that these alternating sums provide additional information to conventional similarity comparisons and power spectrum approaches. We also verified that the Discrete Fourier transform of the complete alternating sum series (not shown) leads to a peaked structure at a ‘frequency’ equal to 16,6666 meaning for the sums a 100/6 characteristic periodicity. Distinctive trends have been identified, e specially around the nucleotide region of the spike protein for all variants studied. These emerging variants seem to have only a few mutations, i.e., no more than a dozen amino acid changes out of the 1200 building blocks that make up the spike protein. They give a selective advantage for their replicating capacity [5,18].

We downloaded worldwide sequencing data of coronavirus variants from GISAID and verified multiple deviations from the originating first Wuhan sequences identified over a year ago. Our statistical representation of coronavirus genome variants, taking sums with both signs, can reveal signals from future genome evolution at the level of nucleotide ordering. These observations lie at the heart of future studies. One could investigate further the present numerical results to study the *n*-moment calculations $E(X^{n})$, and the tail of the series, which could uncover even more underlying properties of the sequences of viral strains related to the pathogens of SARS-CoV-2.

## Figures and Tables

**Figure 1 genes-12-00973-f001:**
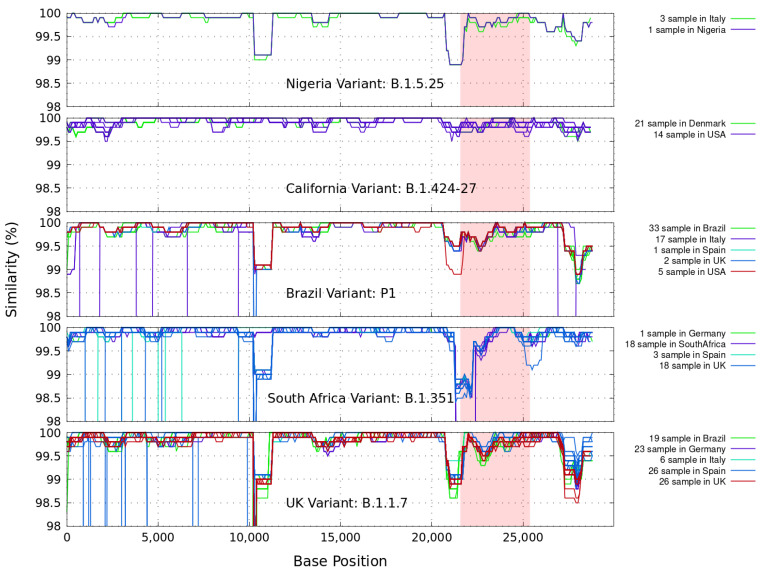
Genetic similarity plot between the query sequence SARS-CoV-2 Wuhan-Hu-1 and several representative full-length genome sequences grouped by variants as found in broad geographical regions, for different numbers of samples, and collected between 01 Nov 2020 and 01 March 2021. A sliding 1000 base pair (bp) window in steps of 100 bp is used. In red is the genomics region encoding the spike (S-protein). The samples shown for a given country correspond to data from a given variant. For example, the difference between the 33 and 19 samples from Brazil means that 33 of the samples are in correspondence with the Brazilian variant P1, and the other different 19 samples are also from Brazil, but they contain the UK variant B.1.1.7.

**Figure 2 genes-12-00973-f002:**
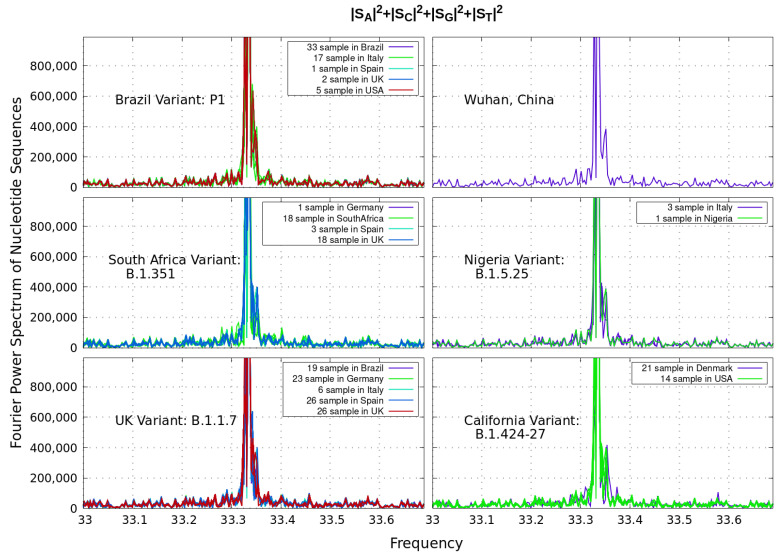
Discrete Fourier Transform identifying base periodicity property of the coronavirus sequence. The plots reveal a distinct peak above noise level at around ‘frequency’ 33.3333 for all variants considered. The differences shown in brackets for a given country correspond to data from a given variant.

**Figure 3 genes-12-00973-f003:**
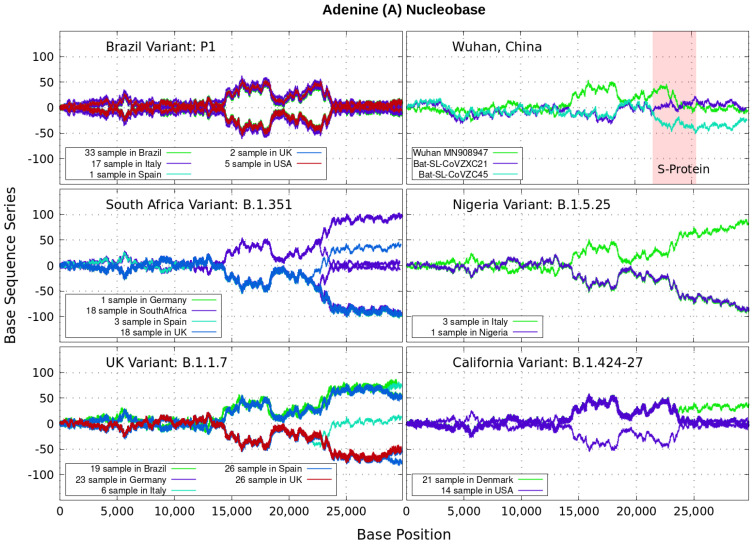
Variant imprints displayed by the nucleotide base: Adenine according to its progression via Equation (Equation 1) along different numbers of samples of the genomics strand of coronavirus available from different countries. In red is the genomics region encoding the spike (S-protein). We also illustrate complete genome sequences of SARS-CoV-2 from the city of Wuhan in China and the bat-derived severe acute respiratory syndrome (SARS)-like coronaviruses: bat-SL-CoVZC45 and bat-SL-CoVZXC21 [1]. *Note*: In all Figure 3, Figure 4, Figure 5 and Figure 6, the differences in samples shown for a given country correspond to data from a given variant. For example, the difference between the 33 and 19 samples from Brazil means that 33 of the samples correspond with the Brazilian variant P1, and the other different 19 samples are also from Brazil, but they contain the UK variant B.1.1.7.

**Figure 4 genes-12-00973-f004:**
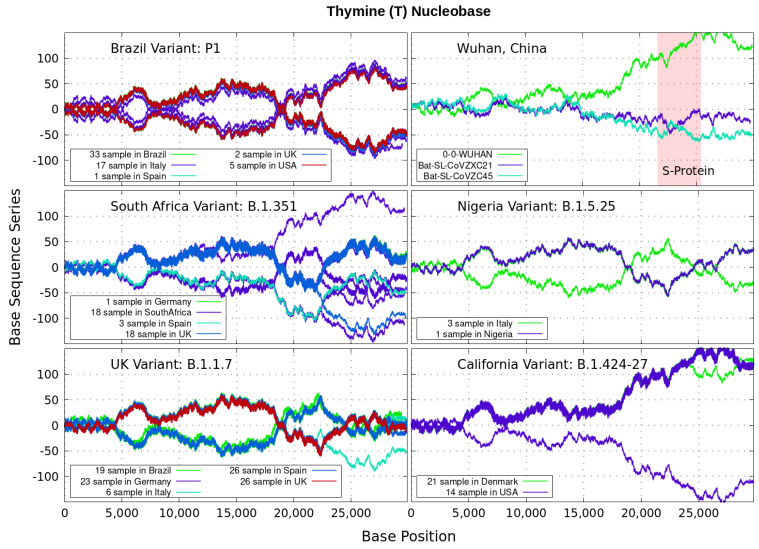
Variant imprints displayed by the nucleotide base: Thymine. *Note*: The coronavirus genome uses RNA for encoding proteins, not DNA. The nucleotide base *T* is related to (U)racil in the nucleic acid RNA via three nucleotide bases, or mRNA codon. This allows to correlate symbols used for nucleic acids (polymers of nucleotides) to that of proteins (polymers of amino acids).

**Figure 5 genes-12-00973-f005:**
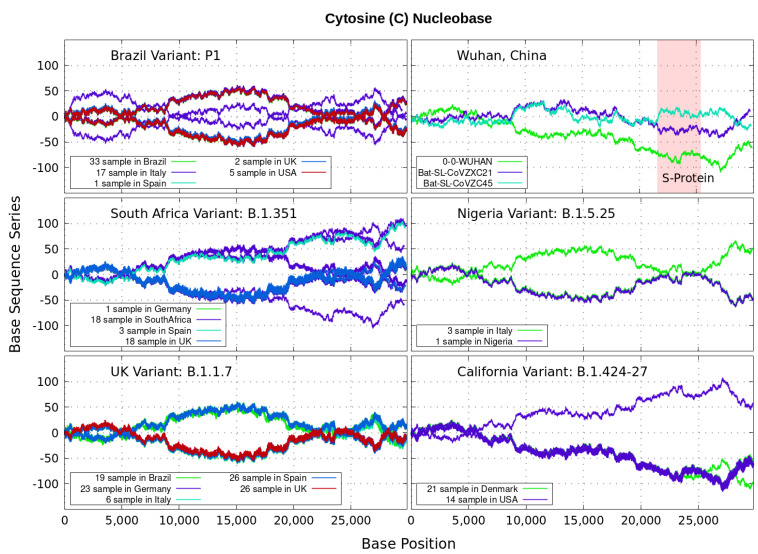
Variant imprints displayed by the nucleotide base: Cytosine.

**Figure 6 genes-12-00973-f006:**
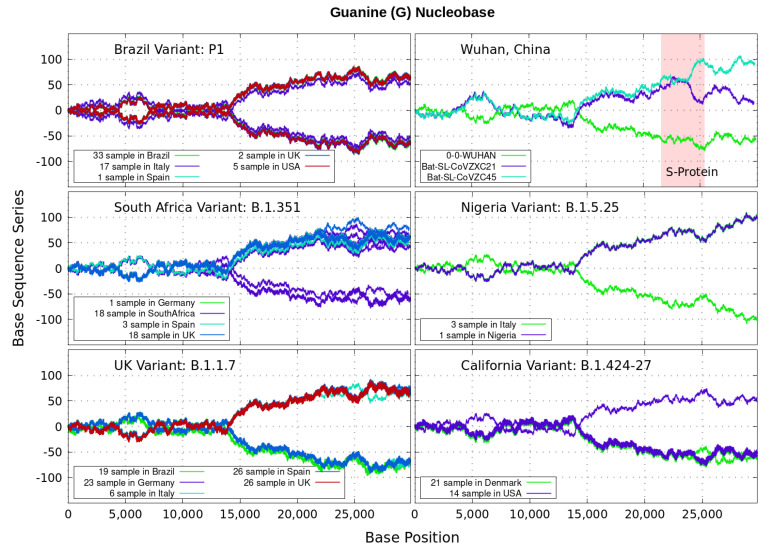
Variant imprints displayed by the nucleotide base: Guanine.

## Data Availability

The accession numbers of the sequences used in the figures are listed in the Appendix A.

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
