# Peer review of "Uncovering Signals from the Coronavirus Genome"

_genes, 2021, doi:10.3390/genes12070973_

Round 1
Reviewer 1 Report
Canessa in the paper “Uncovering signals from the coronavirus genome” proposes a signal analysis method that could help in genome variants surveillance through the suggestion of a quantitative method to examine the distinctive pattern of the coronavirus genome.
The introduction section is well written and explained the background, the importance of the research, and it is easy to follow. Canessa compared the full genome of the Wuhan virus isolate to different isolates including; B.1.1.7 (UK), B.1.135 (South Africa), B.1.429-B.1.427 (California), B.1.525 (Nigeria), and P1 (Brazil).
Minor comments.
In figure 1, I see the countries are distributed in a hard way to follow. For example: what is the difference between (33) samples from Brazil and (19) samples from Brazil? Same case for other countries. Does it mean, for example, the Brazilian variant unique to 33 samples but not 19? Maybe adding some numeration for each plot will help in explaining
Is the script used for generating the plots are publicly available?
Also, for the other figures, it would be great if the author can number them and explain them briefly
Thanks
Author Response
Manuscript ID: genes-1224297
Title: Uncovering signals from the coronavirus genome
Authors: E Canessa
REPLY to Reviewer 1:
The author would like to thank this anonymous Referee Nr.1 for his/her positive comments and suggested minor corrections. This Reviewer understood very well the description and originality of our proposed characterisation of genomes and its potentialities for a new signal analysis of recent sequenced genoma for coronavirus variants.
The Referee acknowledges that the introduction provides sufficient background and includes all relevant reference, and that our results are clearly presented. In his/her words: "The introduction section is well written and explained the background, the importance of the research, and it is easy to follow".
Since in Figures the space left to add further numbering is already extremely reduced, now we have better clarified in the text, and in the Figures captions, that -for example- the difference between the (33) and (19) samples from Brazil in Figs.1-6 means that (33) of the samples taken from GISAID are in correspondence with the Brazil variant P1, and the other different (19) samples are also from Brazil but they contain the UK variant B.1.1.7
We are developing and documenting the open version of the source code (scripts) used for generating our plots. We will publish it and make it publicly available (under the name "GenomaBits") in a follow up paper. Work is in progress.
English improvements and spelling on the manuscript have been made as required.
The author acknowledges again this Reviewer.
-----------------------------
Reviewer 2 Report
In this study, E. Canessa utilize Discrete Fourier transformation and to identify patterns in the genomes of coronavirus variants available in the GISAID database. The author's data show that such an analysis can predict signals from evolutionary events occurring in the future. The manuscript needs improvement in the following areas.
- Introduction to coronavirus genome organization is needed.
- The research question asked is very vaguely stated in the introduction and should be clearly stated.
- The author assumes that conventional Watson:Crick base pairing solely drives genome organization and structure; this is not true for RNA viruses. Non-canonical base pariging, secondary stem loops, knots are also crucial to genome organization as are interactions with virus encoded and host proteins. Thus, the inherent assumption in the study is too simplistic.
- Viral evolution is driven not only by sequence but also by the host immune response to infection. Immunocompromised hosts / therapy pressure may encourage rapid evolution of minor variants that can then successfully populate. There are multiple additional factors such as gender, age, diet, microbiome and immune status that affect viral evolution. These factors are not considered to be important by the author and this reduces my enthusiasm for the manuscript substantially.
Author Response
Manuscript ID: genes-1224297
Title: Uncovering signals from the coronavirus genome
Authors: E Canessa
REPLY to Reviewer 2:
The author would like to thank Referee Nr.2 for his/her review.
We note that in our study we deal with a new type of finite alternating sum series having independently distributed terms associated with binary (0,1) indicators for the nucleotide bases. We show that our original method provides additional information to alternative conventional Similarity comparisons and Discrete Fourier transformation approaches.
As described in detail in the manuscript, using our signal analysis method differs from Discrete Fourier transformation.
We compared the full genome of the Wuhan virus isolate to different other isolates, including: B.1.1.7 (UK), B.1.135 (South Africa), B.1.429-B.1.427 (California), B.1.525 (Nigeria), and P1 (Brazil). Our new method could help in genome variants surveillance through the suggestion of a quantitative method to examine the distinctive pattern of the coronavirus genome.
-----------------------
Round 2
Reviewer 2 Report
The manuscript still needs a substantial amount of language editing in terms of grammar and language. Introduction to the subject is still very haphazard. The author should introduce DTFT and the advantages it provides over sequence homology methods.
Author Response
Manuscript ID: genes-1224297
Title: Uncovering signals from the coronavirus genome
Authors: E Canessa
REPLY to Reviewer 2 bis:
The Referee acknowledges that the research design is appropriate, the methods are adequately described, our results are clearly presented and that our conclusions are supported by the results. English has been improved.
We provide now in the introduction sufficient background and include all relevant references. Further introduced are DTFT and sequence homology methods.
The author would like to thank again Referee Nr.2 for his/her second review.
-------------
